# Long-term visual field evaluation of trabeculectomy patients with a mean intraocular pressure below 15 mmHg

Marina Rocha[1]*, Jayter Silva Paula[2], Cassia Senger[2,3], Tiago Santos Prata[4,5,6,7], Fábio Nishimura Kanadani[4,5,6], Bruno de Barros Massote[4], Vitor Porto de Souza[7], Ana Cláudia Alves Pereira[8], Marcos P. Ávila[1], Leopoldo Magacho[1,9]

1 Ophthalmology Department, Federal University of Goiás, CEROF-HC-UFG, Goiânia, Brazil, 2 Department of Ophthalmology, Ribeirão Preto Medical School, University of São Paulo, São Paulo, Brazil, 3 Bauru Medical School, University of São Paulo, Bauru, Brazil, 4 Ophthalmological Center of Minas Gerais (COMG), Belo Horizonte, Brazil, 5 Department of Ophthalmology, Federal University of São Paulo, São Paulo, Brazil, 6 Department of Ophthalmology, Mayo Clinic, Jacksonville, Florida, United States of America, 7 Glaucoma Unit, Opty Group Brazil, São Paulo, Brazil, 8 Eye Hospital of Mato Grosso do Sul and Federal University of Mato Grosso do Sul, Campo Grande, Brazil, 9 VER Hospital, Goiânia, Brazil

☉ These authors contributed equally to this work.
* dramarinaoftalmo@gmail.com

## Abstract

The purpose of this multicenter, longitudinal, retrospective, observational study was to evaluate the long-term visual field (VF) outcomes of glaucoma patients with a mean intraocular pressure (IOP) of less than 15 mmHg who underwent trabeculectomy (TRAB). Patients who underwent TRAB with at least 3 years of follow-up were divided into two groups: G1 (mean postoperative IOP between 6 and 9 mmHg) and G2 (mean postoperative IOP between 10 and 14 mmHg). A total of 49 eyes from 47 patients were included in G1, and 111 eyes from 90 patients were included in G2. The mean follow-up time was $4.72 \pm 2.72$ years in G1 and $4.63 \pm 2.46$ years in G2 (p = 0.832). The mean IOP was $7.88 \pm 1.06$ mmHg in G1 and $11.83 \pm 1.41$ mmHg in G2 (p < 0.001). The mean progression rate according to the mean deviation (MD) of the VF was $-0.25 \pm 1.09$ dB/year in G1 and $-0.27 \pm 1.16$ dB/year in G2 (p = 0.786). The changes in the visual acuity or VF indices over time were nonsignificant between the groups (p > 0.05 for all). TRAB is a safe procedure with mean IOPs either in the single digits (i.e., between 6 and 9 mmHg) or between 10 and 14 mmHg.

## Introduction

Glaucoma, a group of diseases characterized by progressive neuropathy of the optic nerve and/or the nerve fiber layer of the retina, is the main cause of irreversible blindness worldwide [1]. To date, only one modifiable risk factor for this disease has been

**Data availability statement:** All relevant data are within the paper and its Supporting Information files.

**Funding:** No funding was received at this time for the research. If accepted, the publication fees will be provided by the research incentive fund from the Eye Hospital of the Federal University of Goiás, Goiânia - GO, Brazil.

**Competing interests:** The authors have declared that no competing interests exist.

identified: intraocular pressure (IOP) [2–4]. Several studies have shown that reducing IOP delays or even prevents the progression of glaucoma [5,6].

In eyes with advanced glaucoma, surgery may be necessary, particularly for establishing a target IOP equal to or less than 14 mmHg with little variability [7,8], which is often difficult to achieve with only medical treatment [9]. Among patients with advanced glaucoma, trabeculectomy (TRAB) has also been suggested to yield better outcomes than medical treatment [8].

The target IOP for each patient can vary greatly, and in specific cases, such as for patients with disease progression despite an "adequate" IOP, an IOP below 10 mmHg may be desired, a so-called "single-digit IOP" that lies outside the hypotonia range (IOP ≤ 5 mmHg) [8–10]. However, limited evidence exists regarding whether there are significant differences in terms of preventing or even reducing the rate of progression of glaucoma between patients with a reduced IOP within the target range and those with single-digit IOPs.

In a previous study, Lee et al [11]. reported that IOP was not a predictive factor for eyes with a mean IOP < 15 mmHg and presenting with rapid thinning of the retinal nerve fiber layer (RNFL). However, patients with initial glaucoma and those receiving medical treatment were included in this study. Additionally, the mean IOP was close to the study cutoff point (approximately 13 mmHg) in both groups, and the authors performed an evaluation on the basis of risk factors rather than mean IOP levels.

Furthermore, there are no reports in the literature on the influence of IOP levels within the target range, even for advanced glaucoma, on visual field (VF) evaluations in patients with glaucoma. Thus, the purpose of the present study was to evaluate the long-term VF outcomes of glaucoma patients who underwent TRAB with a mean IOP less than 15 mmHg and to compare these outcomes between patients with an IOP between 6 and 9 mmHg and those with an IOP between 10 and 14 mmHg.

## Materials and methods

This was a multicenter, longitudinal, and observational study. This study was retrospective in nature, and the requirement for informed consent was waived by the Research Ethics Committee (CEP) of the Federal University of Goiás (UFG), which approved it under no. CAAE 59558922.7.1001.5078. This study was performed in accordance with the Declaration of Helsinki. Data were collected from September 1 to December 31, 2022. Since the data source is medical records, researchers had access to information that could identify research participants. All patients with glaucoma who underwent TRAB performed by any of the authors with at least 3 years of postoperative follow-up were retrospectively considered.

A post hoc sample size calculation was performed based on the 2 × 3 factorial ANOVA model (group × severity) used to assess the functional progression rate. A moderate effect size (f = 0.25), an alpha error of 5% (α = 0.05), and a statistical power of 80% (1 − β = 0.80) were considered. The calculation was conducted in Python using the FTestAnovaPower function from the statsmodels library, which indicated that 128 eyes would be required to ensure adequate power. Since this is

a retrospective study, the power analysis was performed only to assess the sensitivity of the available sample and not to guide recruitment.

One or both eyes of patients aged at least 18 years at the time of TRAB were included in this study. All patients had their best-corrected visual acuity (BCVA) recorded preoperatively (up to 3 months before surgery) and on the date of the last visit. Visual acuity was converted to the logarithm of the minimum angle of resolution (LogMAR) for analysis. For inclusion in the study, an examination of the optic disc showing a lesion typical of the disease [12] and at least 1 VF test with the SITA Standard 24−2 protocol (Humphrey Systems, Dublin, CA, USA) revealing characteristic glaucoma changes [13] in the preoperative period were needed. For patients who underwent phacoemulsification and trabeculectomy surgery (Phaco-TRAB), the "preoperative" VF mean deviation (MD) was that first recorded between discharge from surgery to 8 months after the date of surgery. The "preoperative" VA was the value recorded at surgical discharge. The eyes were also compared according to the severity of glaucoma based on the preoperative MD: initial glaucoma (MD > −6.00 dB), moderate glaucoma (−12.00 dB ≤ DM ≤ −6.00 dB) or advanced glaucoma (MD < −12.00 dB) [13].

Patients who had a history of any surgical complications that could affect surgical outcomes, or any other cause that could interfere with the assessment of VA and/or VF examination were excluded. However, if a patient had a history of serious complications, such as visual loss, after at least 3 years of follow-up after TRAB, the last visit before the date of the complication was considered the end of follow-up. Eyes for which the TRAB procedure had been considered to have failed within less than 3 years that underwent successful needling [14] or more than one TRAB and met the inclusion criteria, the date of this last procedure, rather than the date of TRAB, was considered the starting point for data collection. On the other hand, the need for any other surgery was considered a failure criterion.

Patients were evaluated at three time points (preoperative, "mid-follow-up" and last follow-up). When the follow-up time was less than 5 years, the "mid-follow-up time" was the time of the evaluation prior to the final follow-up, which was performed at least 1 year from both the initial evaluation and the final evaluation. The progression rate was calculated as the ratio of the change in the MD (final MD – initial MD) to the follow-up time and was classified as stable/slow progression (positive values to −1.00 dB/year), moderate progression (from − 1.01 to −2 dB/year) or fast progression (greater than −2 dB/year).

Patients were divided into two groups based on their mean IOP during follow-up: Group 1 (G1) included patients with a mean IOP between 6–9 mmHg, and Group 2 (G2) included those with a mean IOP between 10–14 mmHg, with or without the use of medication. The upper threshold of 14 mmHg was selected because it is usually accepted as the upper level of the target IOP for advanced glaucoma [15,16]. The chosen threshold of 6–9 mmHg IOP corresponds to a single-digit IOP, which is often achieved after successful trabeculectomy while also being outside the range considered hypotonic [8,9]. A single-digit IOP may be required not only for normal-pressure glaucoma eyes, but also for selected glaucoma patients progressing despite IOP within the target [10].

The IOPs were measured with the same Goldmann tonometer calibrated at each center responsible for the surgery by trained examiners. For a patient to be included, only 20% of the IOP readings could be above 15 mmHg or below 6 mmHg. However, any IOP measurement above 21 mmHg or below 5 mmHg was considered an exclusion condition. The IOP readings obtained from the date of surgery to surgical discharge were not considered for the study.

The normality of the data was assessed with the Kolmogorov–Smirnov test. The preoperative profiles were compared between the groups with Pearson's chi-square test or the independent Student's t test. Parameters were compared between the preoperative period and the final evaluation with the Friedman ANOVA followed by Bonferroni-corrected pairwise comparisons. Absolute and percentage changes were calculated between the preoperative period and the final evaluation for between-group (G1 vs. G2) and among-severity (initial, moderate and advanced glaucoma) comparisons with the Mann–Whitney U and Kruskal–Wallis tests followed by the Nemenyi *post hoc* test, respectively. Spearman's correlation analysis was used to evaluate the relationship between the rate of progression and the mean IOP in the groups. The trends of variation in the MD and visual field index (VFI) were estimated via Poisson regression with robust variance

as a function of the patients' follow-up times. All 160 eyes had baseline and final visual field examinations available for analysis. However, only 73 eyes had intermediate visual field data obtained between 3 and 5 years of follow-up. Analyses were performed using all available data without imputation for missing intermediate tests. Patients with at least baseline and final assessments were included in the longitudinal analysis. Using eyes with fewer visual fields may have underestimated variability in progression rates, which in turn could have affected the stability estimate.

The data were analyzed with the Statistical Package for the Social Science (IBM Corporation, Armonk, USA) version 26.0. A 5% significance level was adopted for all tests ($p < 0.05$).

## Results

A total of 160 eyes of 126 patients were included in this study: 49 eyes of 47 patients in G1 and 111 eyes of 90 patients in G2. The characteristics of the patients are illustrated in Table 1. Four eyes were excluded due to low IOP (IOP < 6 mmHg

**Table 1. Characterization of the patients in the two groups ($n = 160$).**

| | Groups | | $p^*$ |
| --- | --- | --- | --- |
| | G1 | G2 | |
| **Follow-up time – years** | 4.72 ± 2.72 | 4.63 ± 2.46 | 0.832 |
| **Age at inclusion – years** | 61.31 ± 15.03 | 60.20 ± 12.61 | 0.630 |
| **Sex, n (%)** | | | |
| Female | 26 (53.1) | 55 (49.5) | 0.689 |
| Male | 23 (46.9) | 56 (50.5) | |
| **Eye, n (%)** | | | |
| Right | 25 (51.0) | 52 (46.8) | 0.626 |
| Left | 24 (49.0) | 59 (53.2) | |
| **MD (dB)** | −15.54 ± 9.10 | −10.82 ± 8.19 | **0.001** |
| **Severity by MD, n (%)** | | | |
| Initial | 10 (20.4) | 38 (34.2)≠ | **0.012** |
| Moderate | 7 (14.3) | 35 (31.5)≠ | **0.001** |
| Advanced | 32 (65.3)≠ | 38 (34.2) | **0.023** |
| **Diagnosis, n (%)** | | | |
| Congenital | 0 (0.0) | 1 (0.9) | 0.826 |
| POAG | 42 (85.7) | 91 (82.0) | |
| CAG | 2 (4.1) | 10 (9.0) | |
| Juvenile | 1 (2.0) | 3 (2.7) | |
| Pigmentary | 1 (2.0) | 2 (1.8) | |
| Secondary | 3 (6.1) | 4 (3.6) | |
| **Surgical technique, n (%)** | | | |
| Needling | 3 (6.1) | 4 (3.6) | 0.766 |
| Phaco-TRAB | 12 (24.5) | 29 (26.1) | |
| TRAB | 34 (69.4) | 78 (70.3) | |
| **Mean IOP (mmHg)** | 7.88 ± 1.06 | 11.83 ± 1.41 | **<0.001** |
| **Preop BCVA (LogMar)** | 0.41 ± 0.29 | 0.36 ± 0.30 | 0.363 |
| **Preop PSD (dB)** | 7.13 ± 3.80 | 5.99 ± 3.68 | 0.079 |
| **Preop VFI (%)** | 65.00 ± 26.93 | 74.93 ± 27.94 | 0.150 |

*Chi-square; ≠ Post hoc; n, number; %, percentage; Standard deviation (SD); Mean Deviation (MD); Primary Open Angle Glaucoma (POAG); Closed Angle Glaucoma (CAG); Phacoemulsification and Trabeculectomy (Phaco-TRAB); Intraocular Pressure (IOP); Best-Corrected Visual Acuity (BCVA); Pattern Standard Deviation (PSD); Visual Field Index (VFI).

more than 20% of the time). Fig 1 illustrates the correlation between the progression rate and the mean IOP obtained throughout the study, which was grouped according to the severity of glaucoma in each group. All the correlations were weak and not significant. Fig 2 shows the comparison of VF progression rates between G1 and G2 as a function of glaucoma severity (factorial ANOVA). No difference was observed when G1 and G2 were compared (p = 0.81) or according to the severity of glaucoma (p = 0.53) or when the influence of glaucoma severity was evaluated in the group comparison (p = 0.68). Fig 3 illustrate the Poisson regression analyses used to assess the variability in the MD as a function of time (between the first evaluation and the final follow-up, including visits between these times) in the groups. Fig 4 shows the comparison of the variability in the VFI between G1 and G2 as a function of glaucoma severity. The results remained non-significant in most comparisons, except G2 with initial glaucoma (p < 0.01).

In Fig 1, each point represents a sample, and the lines represent the changes across the different evaluations. Although the slope of the line suggests variations in the trend, the Spearman's correlation coefficient (r) and the p value indicate that there was no significant trend in any of the correlations studied. In Fig 2, the factorial ANOVA test was used because two independent variables (glaucoma severity and group [G1 and G2]) were assessed as a function of a dependent variable (rate of progression).

Poisson regression analysis between the MD and VFI by time of evaluation revealed a regression gradient (β) close to zero, that is, it was nonsignificant for both variables, suggesting that there was no contribution of time to the changes in

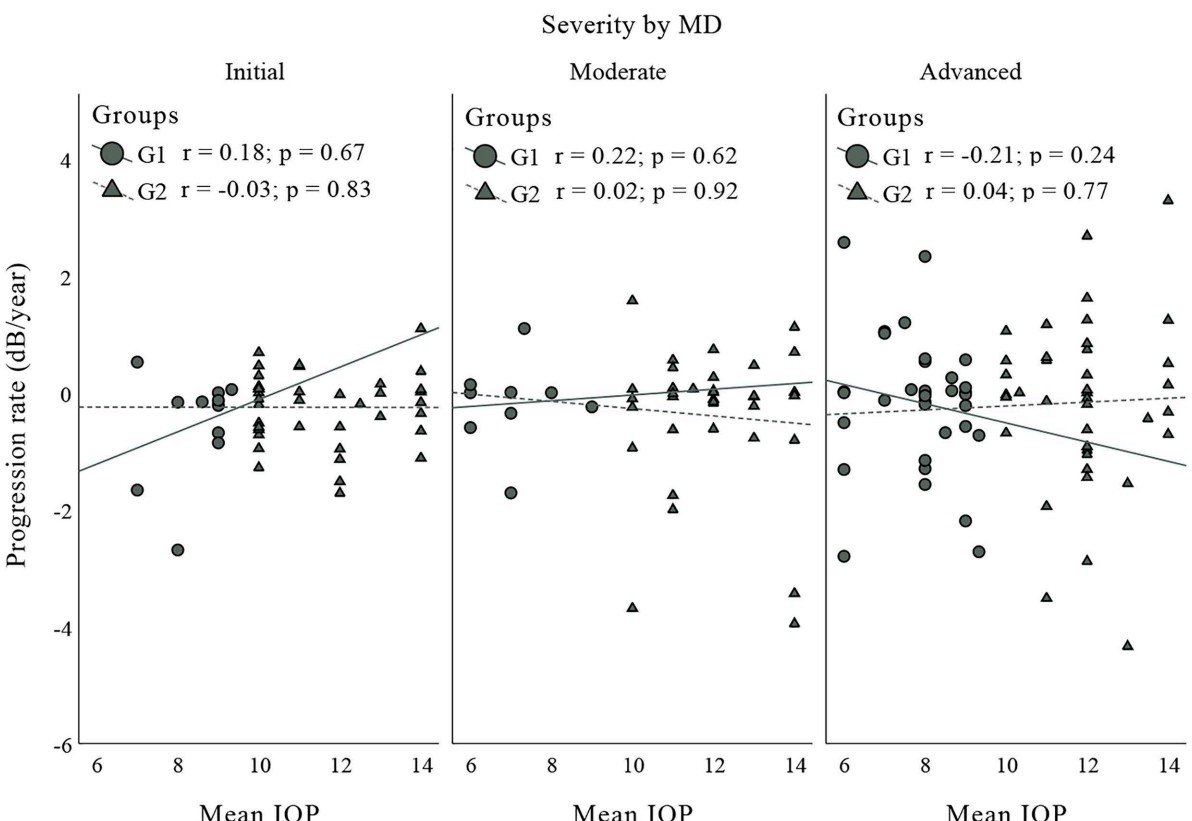

**Fig 1. Scatter plot of the correlation between the annual progression rate and the mean IOP in the groups.**

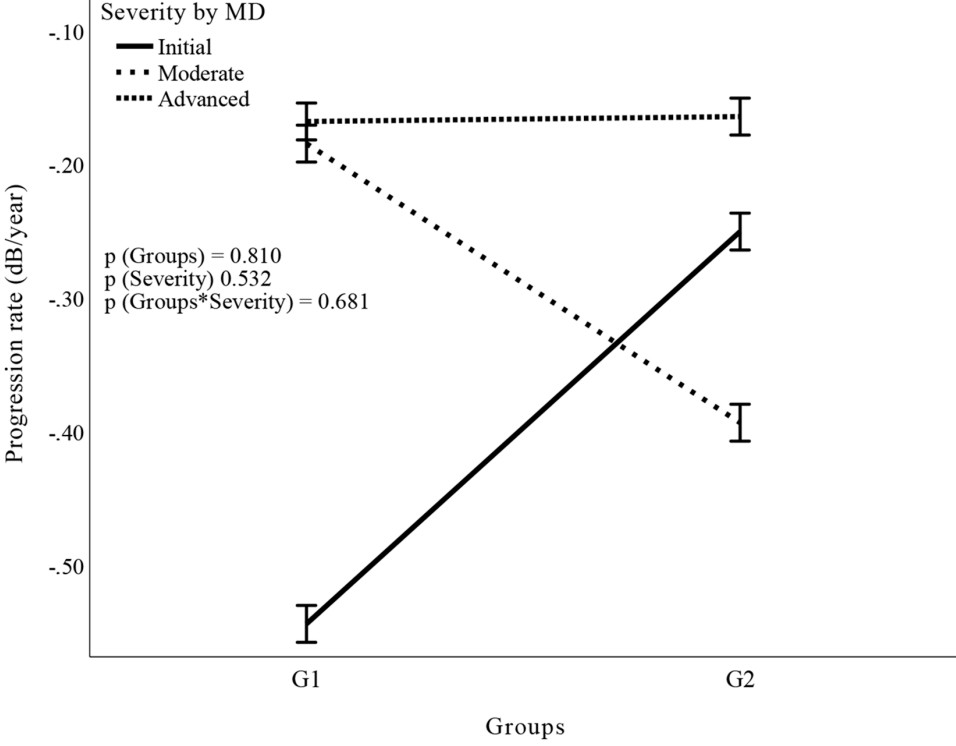

**Fig 2. Results of the factorial analysis of the progression rate in the groups as a function of severity according to the MD.**

the MD and VFI. The line indicates the trend of the model for each group, and despite having a slight slope, the trend is essentially horizontal; that is, there was no significant variation in either variable over time in either group.

## Discussion

The present results suggest that regardless of the glaucoma severity, eyes that maintained a mean IOP between 6 and 9 mmHg in the postoperative period had a similar evolution to those that maintained a mean IOP between 10 and 14 mmHg. Furthermore, in terms of the number of complications, results of visual acuity and VF examinations, and progression rates, the group with the strictest IOP control was not superior to the group with the higher range of pressure values.

Koenig et al. [17] followed 80 eyes of 74 patients who underwent TRAB (61.3% POAG), for a period of 1 year. The mean corrected VA was 0.2 before surgery and remained relatively unchanged at the end of follow-up. After 1 year, the mean IOP was 10 mmHg, and the mean progression rate after surgery was −0.33 dB/year. Despite the short follow-up time, the sample was similar to that of the present study, in which a low annual progression rate was also observed (−0.25 ± 1.09 dB/year in G1 and −0.27 ± 1.16 dB/year in G2) in eyes with a mean IOP below 15 mmHg.

Naito et al. [18] demonstrated reduced progression in eyes with normal-pressure glaucoma and a single-digit IOP after TRAB but with a greater occurrence (17.6%) of reduction in visual acuity in eyes with a mean IOP of 6.0 ± 1.0 vs. 8.6 ± 3.0 mmHg. In the present study, progression to hypotony could not be accurately evaluated because patients were automatically excluded if they entered this evaluation range, unless it occurred after 3 years of follow-up.

A total of 65.3% of the eyes in G1 and 34.23% in G2 were classified as having advanced glaucoma. Several studies have suggested that patients with advanced glaucoma have a greater tendency to experience disease progression, requiring a lower IOP level, often in single digits [8–10]. This could have generated a bias toward a higher rate of

progression, which was not found in the analysis of the entire study population. However, a subanalysis based on glaucoma severity was also performed, which revealed a similar number of eyes with advanced glaucoma in each group (32 in G1 vs. 38 in G2). The subanalysis also revealed similar changes in terms of visual field stability and number of complications. These results were corroborated by the correlation between the annual progression rate and the mean IOP in the groups in relation to glaucoma severity (Fig 1) and by factor analysis (ANOVA) of the progression rate across the groups as a function of severity according to the MD (Fig 2).

The severity of glaucoma had no apparent direct influence on visual field stability during the present study. We chose to perform this analysis only with the MD and VFI, as the PSD shows an increasing trend in glaucoma patients up to a certain severity, after which it usually begins to decrease in more advanced eyes, therefore, analysis of the PSD could lead to unreliable results [13].

There was a trend toward stability during follow-up in both groups, and both groups had similar frequencies of complications. The results of the regression analyses revealed that there was no significant difference in the VFI or MD over time between the groups; that is, there was a tendency toward visual field stabilization. This finding indicates that the postoperative follow-up results for eyes with "less rigid" control of pressure still within what is considered the "safe" pressure range [7,18] were not inferior to those of eyes that achieved greater IOP control.

The mean progression rate in terms of MD was −0.25 ± 1.09 dB/year in G1 and −0.27 ± 1.16 dB/year in G2 (p = 0.786); additionally, 81.6% of the eyes in G1 and 83.8% of the eyes in G2 were classified as slow progressors. This result is similar to that reported by Jammal et al., [2] in which the mean progression rate was −0.20 ± 0.39 dB/year in a population of 1271 POAG patients followed up for 4.6 ± 2.2 years. Shin et al. [19] evaluated a cohort of 127 eyes with advanced POAG for 11 years and reported an overall rate of progression of −0.43 dB/year, with rates of −0.67 dB/year for patients with progressive POAG and −0.20 dB/year for patients with nonprogressive POAG. These findings indicate that the results of the present study are in agreement with those of previous evaluations in the literature.

This study included data from multiple centers, which allowed the documentation of a significant number of eyes with long-term follow-up in the postoperative period. We chose to include only surgical patients and those who required additional TRAB (or needling) because of the greater likelihood of evaluating patients with a mean IOP below 15 mmHg, in addition to maintaining equivalence regarding the potential complications inherent to the surgery itself [20].

The present study aimed to represent the postoperative evolution of these patients in actual clinical practice, or "real life". Several limitations are naturally expected, including that there was no equivalence regarding the number of VF exams per patient over time. Although the patients had a long follow-up period of more than 3 years, many patients had only 2 VF exams, which limited the ability to assess disease progression over time. This is a study bias. However, Poisson regression analysis was performed for the MD and VFI values of most patients who underwent several examinations during the follow-up period to reduce the impact of this important limitation in the analysis of the results, since for both field indices of visual impairment as a function of time, no progression was observed in the study population. In addition, although the majority of patients underwent serial OCT exams, there was no standardization regarding OCT technologies among the centers participating in the study, which precluded a suitably reliable comparison [21].

Imbalances in glaucoma severity at baseline between the two IOP groups raise concerns of confounding, as more advanced cases are concentrated in the lower IOP group (G1). However, this limitation was addressed in the analysis: Fig 1 shows the correlation between the annual progression rate and mean IOP stratified by severity levels. In Fig 2, a factorial ANOVA model was used to assess the main effects of severity, group, and their interaction on the progression rate. None of these comparisons were statistically significant (p > 0.5 for all effects). Fig 3 demonstrates that the delta MD (final – baseline value), already adjusted for baseline severity, did not differ between groups. Finally, Fig 4 presents separate regressions by group and severity level, confirming that baseline heterogeneity did not impact the results, thus supporting the robustness of the findings.

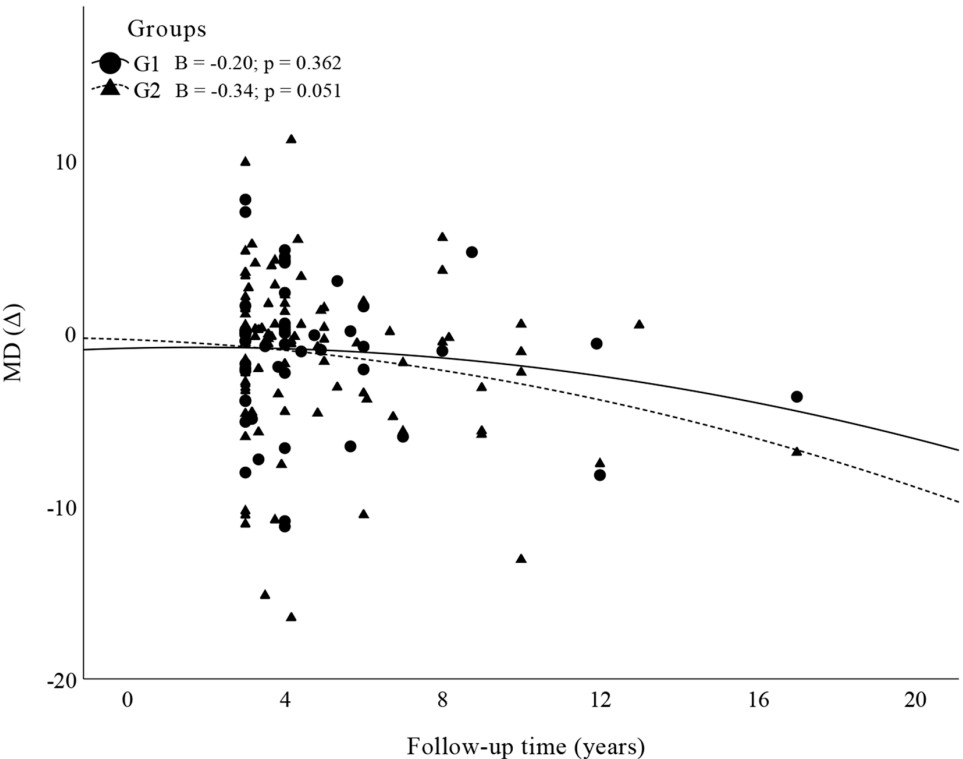

**Fig 3. Results of the Poisson regression analysis of the MD as a function of follow-up time.**

The retrospective nature of the present study, which led to systematic bias, is another limitation. Many patients who developed hypotonia were automatically excluded from the initial analysis. In addition, the surgical success rate and the analysis of TRAB survival were not among the objectives of this study; this may have affected the number of complications observed, as only eyes where the surgeries were successful in terms of pressure reduction were included because the main goal of the evaluation was to determine the influence of reduced IOP levels on VF stability. We did not restrict the period for patient inclusion or access to all medical records of all surgeries performed in the centers to achieve more complete inclusion, which may have generated significant inclusion bias. Another source of bias was the possible inclusion of both eyes of the same patient. Individual factors related to progression are artificially duplicated in such cases. However, as the objective of this study was not to evaluate the effectiveness of surgery, both eyes were included because surgery could exert a distinct influence on each eye of the same patient in terms of the mean IOP measured throughout the follow-up.

To date, the study by Lee et al. [11] has been the only one with a proposal similar to that of the present study. They conducted a retrospective study of 41 eyes of patients with medically treated POAG and a mean IOP less than 15 mmHg who were followed for more than 5 years. Twenty eyes were classified as fast progressors (−0.3 ± −0.4 dB/year), and 21 eyes were classified as slow progressors (0.1 ± −0.3 dB/year) (p = 0.004). Among the eyes evaluated by Lee et al., [11] 50% of fast progressors versus 4.8% of slow progressors presented with disc hemorrhage, indicating that other factors not associated with IOP may play important roles in disease progression. On the other hand, as the patients did not undergo surgical treatment, both groups had similar mean IOP values (approximately 13 mmHg); furthermore, the authors did not evaluate an IOP level below 15 mmHg as a possible factor related to glaucoma progression, which was an assessment that formed the central objective of the present study.

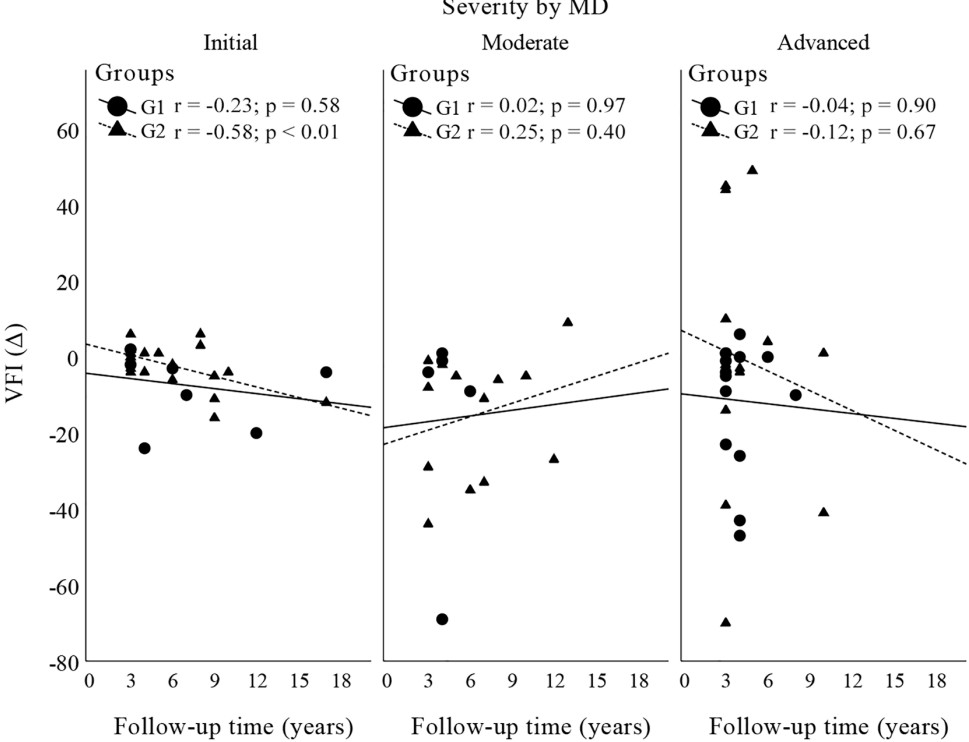

**Fig 4. Scatter plot of the correlation between VFI as a function of follow-up time according to severity.**

Thus, studies evaluating the long-term changes in eyes with glaucoma subdivided into pressure targets in the period following TRAB are lacking. The results presented herein allow us to conclude that the determination of a more rigorous IOP target, in the single digits, may be necessary only in isolated cases where there is progression, even with an IOP apparently within the expected range for a given patient. IOP-independent mechanisms of progression, autoregulatory failure, and/or optic nerve susceptibility may be responsible for glaucoma progression in some, or maybe even the majority, of these patients. Therefore, maintaining a mean IOP below 15 mmHg for patients with glaucoma after long-term TRAB can apparently yield discrete visual stability or the absence of VF loss, regardless of whether the IOP is in the single digits or between 10 and 14 mmHg.

## Supporting information

**S1 File. Stat-Marina2.**
(XLSX)

## Acknowledgments

The authors have no proprietary interest in any of the products mentioned in the text.

## Author contributions

**Conceptualization:** Jayter Silva Paula, Tiago Santos Prata, Fábio Nishimura Kanadani, Ana Cláudia Alves Pereira, Marcos P. Ávila, Leopoldo Magacho.

**Data curation:** Marina Rocha, Cassia Senger, Tiago Santos Prata, Fábio Nishimura Kanadani, Bruno de Barros Massote, Vitor Porto de Souza, Ana Cláudia Alves Pereira, Leopoldo Magacho.

**Formal analysis:** Marina Rocha, Jayter Silva Paula, Tiago Santos Prata, Marcos P. Ávila, Leopoldo Magacho.

**Funding acquisition:** Marcos P. Ávila, Leopoldo Magacho.

**Investigation:** Jayter Silva Paula, Cassia Senger, Tiago Santos Prata, Ana Cláudia Alves Pereira.

**Methodology:** Marina Rocha, Jayter Silva Paula, Tiago Santos Prata, Fábio Nishimura Kanadani, Leopoldo Magacho.

**Project administration:** Marcos P. Ávila, Leopoldo Magacho.

**Supervision:** Jayter Silva Paula, Tiago Santos Prata, Fábio Nishimura Kanadani, Ana Cláudia Alves Pereira, Leopoldo Magacho.

**Validation:** Jayter Silva Paula, Tiago Santos Prata, Fábio Nishimura Kanadani, Ana Cláudia Alves Pereira, Marcos P. Ávila, Leopoldo Magacho.

**Writing – original draft:** Marina Rocha, Cassia Senger, Bruno de Barros Massote, Vitor Porto de Souza.

**Writing – review & editing:** Marina Rocha, Jayter Silva Paula, Leopoldo Magacho.

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
