## [Decision Letter · Decision Letter 0]

6 Oct 2025

Dear Dr. Rocha,

plosone@plos.org. . . . A rebuttal letter that responds to each point raised by the academic editor and reviewer(s). You should upload this letter as a separate file labeled 'Response to Reviewers'.A marked-up copy of your manuscript that highlights changes made to the original version. You should upload this as a separate file labeled 'Revised Manuscript with Track Changes'.An unmarked version of your revised paper without tracked changes. You should upload this as a separate file labeled 'Manuscript'.

We look forward to receiving your revised manuscript.

Kind regards,

Natasha Gautam, MBBS, MS

Academic Editor

PLOS ONE

Journal Requirements:

https://journals.plos.org/plosone/s/file?id=ba62/PLOSOne_formatting_sample_title_authors_affiliations.pdf....

“We would like to thank the support from the research incentive fund provided by CEROF/UFG (Eye Hospital of the Federal University of Goiás, Goiânia-GO, Brazil) for the publication fees.”

“No funding was received at this time for the research. If accepted, the publication fees will be provided by the research incentive fund from the Eye Hospital of the Federal University of Goiás, Goiânia - GO, Brazil.”

Reviewers' comments:

Reviewer's Responses to Questions

**Comments to the Author**

1. Is the manuscript technically sound, and do the data support the conclusions?

Reviewer #1: Partly

Reviewer #2: Yes

2. Has the statistical analysis been performed appropriately and rigorously?

Reviewer #1: No

Reviewer #2: Yes

3. Have the authors made all data underlying the findings in their manuscript fully available?

Reviewer #1: Yes

Reviewer #2: Yes

4. Is the manuscript presented in an intelligible fashion and written in standard English?

Reviewer #1: No

Reviewer #2: Yes

Reviewer #1: While the manuscript addresses an important clinical question regarding long-term visual field stability after trabeculectomy in patients with well-controlled IOP, several methodological limitations raise concerns about the validity of the conclusions.

Imbalances in glaucoma severity at baseline between the two IOP groups raise concerns of confounding, particularly since the more advanced cases are concentrated in the lower IOP group (G1). There is no statistical adjustment (e.g., covariate adjustment) to account for this imbalance. Moreover, the claim is based on non-significant changes in MD and VFI, but absence of statistical significance does not imply stability, especially in a study with variable VF testing intervals and limited test repetitions per patient.

In the discussion section the statement 'This finding indicates that the postoperative follow-up results for eyes with “less rigid” control of pressure still within what is considered the “safe” pressure range were not inferior to those of eyes that achieved greater IOP control.' seems overstated. The study was not powered or designed to show non-inferiority. This language may mislead the reader into thinking the higher IOP group is equally effective, which overgeneralizes the results. Patients with IOP <6 mmHg more than 20% of the time were excluded, but how many were excluded and how that impacted group size or bias is not reported.

Reviewer #2: This study addresses an important clinical question: whether achieving a single-digit IOP after trab confers superior long-term visual field outcomes compared to maintaining IOP in the low-teens. The study is well-structured, with long follow-up, and appropriate statistical methods. However, there are a few concerns:

1. The choice of thresholds (6–9 vs. 10–14 mmHg) is justified clinically, but a rationale (e.g., based on prior studies or consensus guidelines) needs to be stated.

2. Handling of missing data needs to be described.

3. The imbalance in baseline MD and glaucoma severity between the two groups may confound results and should be discussed as a limitation.

4. The discussion should be more concise, and lengthy descriptions of figures (e.g., lines, slopes, r values) are better suited for results section, not discussion.

5. Could expand on clinical implications: What does this mean for glaucoma specialists in practice? Should surgeons avoid aiming for overly aggressive IOP lowering, or is there a patient subgroup that might still benefit?

.

Reviewer #1: No

Reviewer #2: No

---

## [Author Response · Author response to Decision Letter 1]

30 Oct 2025

Dear Editor and Reviewers, we sincerely thank you for the valuable comments and suggestions. A detailed point-by-point response to all comments has been provided in the uploaded file titled "Response to reviewers".

---

## [Decision Letter · Decision Letter 1]

21 Nov 2025

Dear Dr. Rocha,

We look forward to receiving your revised manuscript.

Kind regards,

Natasha Gautam, MBBS, MS

Academic Editor

PLOS ONE

Journal Requirements:

Reviewers' comments:

Reviewer's Responses to Questions

Reviewer #1: (No Response)

Reviewer #2: All comments have been addressed

2. Is the manuscript technically sound, and do the data support the conclusions?

Reviewer #1: Partly

Reviewer #2: Yes

3. Has the statistical analysis been performed appropriately and rigorously?

Reviewer #1: I Don't Know

Reviewer #2: Yes

4. Have the authors made all data underlying the findings in their manuscript fully available?

Reviewer #1: Yes

Reviewer #2: Yes

5. Is the manuscript presented in an intelligible fashion and written in standard English?

Reviewer #1: No

Reviewer #2: Yes

Reviewer #1: Thank you for addressing the earlier concerns and incorporating necessary clarifications into the Methods and Discussion. The added explanations regarding sample size calculation, grouping thresholds, and handling of VF data have improved transparency. A few minor points still require attention:

The added paragraph explains “adequate power,” but because this is a retrospective dataset with a fixed sample, please clarify that the calculation is exploratory rather than a basis for actual sampling. You now state that analyses used all available data without imputation. Please briefly acknowledge in the Discussion that this may underestimate variability in progression, as eyes with fewer VFs may bias the stability estimate. Moreover, The added severity-imbalance paragraph is thorough but overly long; condensing it to a few sentences would improve readability.

Reviewer #2: (No Response)

Do you want your identity to be public for this peer review? For information about this choice, including consent withdrawal, please see our Privacy Policy..

Reviewer #1: No

Reviewer #2: No

---

## [Author Response · Author response to Decision Letter 2]

1 Dec 2025

Dear Editor,

All revisions have been incorporated into the manuscript and addressed in the "Response to reviewers" document. Once again, we thank you and the reviewers for taking the time and effort dedicated to helping us improve our work.

---

## [Editor Report · Decision Letter 2]

23 Dec 2025

Dear Dr. Rocha,

**The reviewers have recommended publication, but also suggest some minor revisions to your manuscript. Therefore, I invite you to respond to the previous reviewer 1 comments and revise your manuscript.**

https://journals.plos.org/plosone/s/submission-guidelines#loc-laboratory-protocols. Additionally, PLOS ONE offers an option for publishing peer-reviewed Lab Protocol articles, which describe protocols hosted on protocols.io. Read more information on sharing protocols at . Additionally, PLOS ONE offers an option for publishing peer-reviewed Lab Protocol articles, which describe protocols hosted on protocols.io. Read more information on sharing protocols at . Additionally, PLOS ONE offers an option for publishing peer-reviewed Lab Protocol articles, which describe protocols hosted on protocols.io. Read more information on sharing protocols at . Additionally, PLOS ONE offers an option for publishing peer-reviewed Lab Protocol articles, which describe protocols hosted on protocols.io. Read more information on sharing protocols at https://plos.org/protocols?utm_medium=editorial-email&utm_source=authorletters&utm_campaign=protocols....

We look forward to receiving your revised manuscript.

Kind regards,

Natasha Gautam, MBBS, MS

Academic Editor

PLOS One

**Journal Requirements:**

**Additional Editor Comments:**

**Previous reviewer 1 comments:**

Thank you for addressing the earlier concerns and incorporating necessary clarifications into the Methods and Discussion. The added explanations regarding sample size calculation, grouping thresholds, and handling of VF data have improved transparency. A few minor points still require attention:

The added paragraph explains “adequate power,” but because this is a retrospective dataset with a fixed sample, please clarify that the calculation is exploratory rather than a basis for actual sampling. You now state that analyses used all available data without imputation. Please briefly acknowledge in the Discussion that this may underestimate variability in progression, as eyes with fewer VFs may bias the stability estimate. Moreover, The added severity-imbalance paragraph is thorough but overly long; condensing it to a few sentences would improve readability.

---

## [Author Response · Author response to Decision Letter 3]

24 Feb 2026

We thank the Editor and the reviewers for their time and consideration of our manuscript. As requested, a detailed point-by-point Response to Reviewers has been provided as a separate document. We appreciate the opportunity to resubmit our work and look forward to the continuation of the review process.

---

## [Editor Report · Decision Letter 3]

8 Mar 2026

Long-term visual field evaluation of trabeculectomy patients with a mean intraocular pressure below 15 mmHg

PONE-D-25-25183R3

Dear Dr. Rocha,

We’re pleased to inform you that your manuscript has been judged scientifically suitable for publication and will be formally accepted for publication once it meets all outstanding technical requirements.

Kind regards,

Natasha Gautam, MBBS, MS

Academic Editor

PLOS One

Additional Editor Comments (optional):

The authors have appropriately responded to comments
---

## [Editor Report · Acceptance letter]

PONE-D-25-25183R3

PLOS One

Dear Dr. Rocha,

I'm pleased to inform you that your manuscript has been deemed suitable for publication in PLOS One. Congratulations! Your manuscript is now being handed over to our production team.

Kind regards,

on behalf of

Dr. Natasha Gautam

Academic Editor

PLOS One